# Does Potassium (K^+^) Contribute to High-Nitrate (NO_3_^−^) Weakening of a Plant’s Defense System against Necrotrophic Fungi?

**DOI:** 10.3390/ijms232415631

**Published:** 2022-12-09

**Authors:** Anis Limami, Bertrand Hirel, Jérémy Lothier

**Affiliations:** 1Univ Angers, INRAE (Institut National Recherche Agriculture Alimentation), 49000 Angers, France; 2Institut Jean-Pierre Bourgin (IJPB), INRAE-AGRo Paris Tech, CNRS, 75016 Versailles, France

**Keywords:** nitrate, potassium, necrotrophic fungi, plant-pathogen relation, CBL/CIPK, jasmonic acid

## Abstract

In this opinion article, we have analyzed the relevancy of a hypothesis which is based on the idea that in *Arabidopsis thaliana* jasmonic acid, a (JA)-mediated defense system against necrotrophic fungi is weakened when NO_3_^−^ supply is high. Such a hypothesis is based on the fact that when NO_3_^−^ supply is high, it induces an increase in the amount of bioactive ABA which induces the sequestration of the phosphatase ABI2 (PP2C) into the PYR/PYL/RCAR receptor. Consequently, the Ca sensors CBL1/9-CIPK23 are not dephosphorylated by ABI2, thus remaining able to phosphorylate targets such as AtNPF6.3 and AtKAT1, which are NO_3_^−^ and K^+^ transporters, respectively. Therefore, the impact of phosphorylation on the regulation of these two transporters, could (1) reduce NO_3_^−^ influx as in its phosphorylated state AtNPF6.3 shifts to low capacity state and (2) increase K^+^ influx, as in its phosphorylated state KAT1 becomes more active. It is also well known that in roots, K^+^ loading in the xylem and its transport to the shoot is activated in the presence of NO_3_^−^. As such, the enrichment of plant tissues in K^+^ can impair a jasmonic acid (JA) regulatory pathway and the induction of the corresponding biomarkers. The latter are known to be up-regulated under K^+^ deficiency and inhibited when K^+^ is resupplied. We therefore suggest that increased K^+^ uptake and tissue content induced by high NO_3_^−^ supply modifies the JA regulatory pathway, resulting in a weakened JA-mediated plant’s defense system against necrotrophic fungi.

## 1. Introduction

### 1.1. The Dual Function of Nitrate as a Nutrient and Signaling Molecule

Nitrogen (N) is an essential inorganic nutrient to sustain plant development and growth. Unlike ammonium, when it occurs in soil, it is either adsorbed on clay particles or oxidized to nitrate by soil microorganisms. Nitrate (NO_3_^−^) is the main source of N available in the soil for plants. Probably because of its importance in mineral nutrition, plants have evolved two distinct NO_3_^−^ transport and absorption systems in order to adapt to its availability in the soil: a high-affinity, low capacity, transporter system (HATS) that belongs to NRT2 (Nitrate Transporter 2) family and a low-affinity, high capacity, transporter system (LATS) that belongs to a family of proteins formerly named NRT1/PTR (Nitrate Transporter 1/Peptide Transporter), renamed NPF (NRT1/PTR Family) [1].

The function of NO_3_^−^ as a signaling molecule has been studied in depth mostly in the model plant *Arabidospsis thaliana*, and notably in relation to the developmental plasticity of the root system architecture [2,3] that allows an optimal adaptation of the plant-to-soil N availability [2,4,5]. In particular, it has been shown that when there is a high supply of NO_3_^−^, lateral root (LR) elongation is inhibited right after their emergence from the primary root. It has been proposed that such an inhibitory mechanism is under the control of the tissue of NO_3_^−^ content following the finding that in a nitrate reductase-deficient mutant, its systemic inhibitory effect on lateral root growth does not occur [5]. When high rates of NO_3_^−^ were applied locally to a root system subjected to nitrate deficiency, LR growth was locally stimulated and oriented towards the source of nitrate. The localized effect of nitrate on LR growth was shown to be mediated by the nitrate transporter AtNPF6.3 (also known as NRT1.1/CHL1.5) [6]. It has been shown that under a high NO_3_^−^ supply, AtNPF6.3-dependent auxin basipetal transport is inhibited by nitrate, leading to auxin accumulation in the LR tip, which stimulates LR emergence. Conversely, under a low nitrate supply, AtNPF6.3 transports auxin away from the LR tip, thus decreasing its content in the tip, and causing, therefore, the inhibition of LR outgrowth [7].

### 1.2. High-NO_3_^−^ Supply Increases Biotic Stress Susceptibility

In comparison with the above-mentioned roles of nitrate in plant development, the role of this nutrient in plant response to pathogens is less known [8,9]. Still, on the basis of experimental and field experiments, it has been observed that overapplication of N fertilizer may affect plant–pathogen interaction, causing in some cases an enhanced disease susceptibility [10,11]. However, this view should be tempered as it appears that NO_3_^−^ interference with plant defense machinery is more complex and specific to each pathosystem, as disease severity may be linked to either increasing or decreasing N fertilizer [12,13]. Therefore, considering N only from a nutritional angle, i.e., as a precursor of nutrients and defense molecules, seems unreliable [14].

Indeed, more recently, the idea of a signaling role of NO_3_^−^ in plant health emerged and is supported by an increasing amount of evidence [8,9,10,15,16,17,18]. The studies suggest that NO_3_^−^ may interfere as a signal molecule in the signaling pathways that lead in each pathosystem to the production of specific signaling molecules such as salicylic acid, or hormones such as ethylene and jasmonic acid (JA); in turn, these molecules trigger the expression of specific genes belonging to families such as pathogen related (PR) or the antifungal plant defensin family (PDF) [12,13,19,20,21]. In line with these considerations, one of the most relevant findings in our opinion was the discovery of an additional role for the high-affinity nitrate transporter of *Arabidopsis thaliana* AtNRT2.1 as being linked to plant defense against the bacterial pathogen *Pseudomonas syringae* pv tomato DC3000 (Pst) [15,22]. Authors found that a functional AtNRT2.1 antagonizes the priming of the plant’s defense against *Pseudomonas syringae*. Conversely, deletion mutant *nrt2*, in which the priming of salicylic acid signaling operated properly, showed a reduced susceptibility to the pathogen. Very interestingly, in *nrt2*, hormonal homeostasis was concomitantly affected with irregular function of JA and abscisic acid (ABA) pathways upon infection [15,22].

Studies on various pathosystems such as tomato/*Oidium* [23], *Arabidopsis thaliana*/*Botrytis cinerea* [8,9], and *Arabidopsis thaliana*/*Alternaria brassisicola* [24] report that high concentrations of NO_3_^−^ in the growth medium increase plant susceptibility to necrotic fungi. Tests of pathogenicity following *Arabidopsis thaliana* inoculation with *Botrytis cinerea* carried out on plants grown on 0.5, 2, and 10 mM NO_3_^−^ showed that leaf lesion areas (cm^2^) were larger in plants grown on 2 and 10 mM compared to those grown on 0.5 mM NO_3_^−^ [9]. Furthermore, the lesion propagation rate (cm^2^/24 h) was positively correlated to NO_3_^−^ concentration in the medium; for example, the propagation rate of the wild-type strain Bd90 was nil on leaves of plants grown on 0.5 mM NO_3_^−^, increasing to 0.1 at 2 mM NO_3_^−^ and doubling to reach 0.2 at 10 mM NO_3_^−^ [9]. Similarly, susceptibility of *Arabidopsis thaliana* to *Alternaria brassisicola* at the rosette stage was largely higher under nitrate (5 mM) compared to ammonium (5 mM) conditions [24]. The lesion area was tiny on leaves of ammonium-fed plants whereas it was almost 10 times bigger (ca. 0.7 mm^2^ at 7 days’ post-infection) in nitrate-fed plants [24].

After reviewing the literature, we attempted to decipher mechanisms through which NO_3_ supplied at high concentration would cause deleterious effects on plant health. For this aim, we focused on the well-studied model plant *Arabidopsis thaliana* and its interaction with necrotrophic fungi.

## 2. NO_3_^−^ Uptake, Signaling, and Sensing Involves Calcineurin B-Like (CBL)-Interacting Protein Kinase (CIPK)

AtNPF6.3 was shown to interact with other molecules as a component of a macromolecular complex dedicated to nitrate sensing and signaling along with a protein kinase, CIPK23, calcium sensors (CBL1 and CBL9) [25], and ABI2, a phosphatase of the phosphatase 2C family [26]. CIPK23 belongs to a family of protein kinases (calcineurin B-like (CBL)-interacting protein kinase (CIPK)), encompassing 26 members in *Arabidopsis thaliana* [27]. Each CIPK specifically interacts with one or several of the 10 CBL calcium sensors to specifically decode calcium signals [28,29]. Specific roles for some CIPK–CBL pairs have been elucidated and several targets have been identified [30,31,32]. CBL1/9 interact with and activate CIPK23 that, in turn, phosphorylates AtNPF6.3, causing a decrease in its NO_3_^−^ absorption capacity. The phosphatase ABI2, by dephosphorylating CIPK23 and the calcium sensor CBL1, counteracts their action on nitrate transport, signaling, and sensing by allowing AtNPF6.3 to remain in an unphosphorylated state [26].

## 3. High-NO_3_^−^ Supply Induces an Increase in Bioactive ABA Content in Planta

Although deficiencies in essential mineral nutrients, e.g., N, phosphorus (P), and potassium (K), lead to common stress reactions such as an increase in reactive oxygen species, involving ABA can be different depending on the type of the nutrient [33]. Comparison of the changes in ABA concentration in *Ricinus communis* under mineral nutrients deficiencies showed contrasted results between P and N. The ABA concentration increased 29-fold in xylem sap under P limitation whereas it showed a 5-fold increase in the xylem with the increase in NO_3_^−^ supply (see Figures 1 and 2 in [34]). Accordingly, ABA concentration increased in the leaves showing a linear positive correlation between ABA content and NO_3_^−^ supply (Figure 1).

In *Arabidopsis thaliana*, ABA is mobilized for the local stimulation of lateral root elongation by patches of high NO_3_^−^ [35]. Moreover, it has been shown that the effect of nitrate on ABA concentration is controlled by a signaling pathway that resulted in the release of bioactive ABA from the inactive conjugated form, ABA-glucose ester (ABA-GE), by the action of enzymes of the beta-glucosidase (BG) family [36]. This release catalyzed by the enzyme beta-Glucuronidase 1 primarily occurs in the root tips, which then allows it to transduce the NO_3_^−^ mediated hormonal signal to other organs [36].

## 4. High-NO_3_^−^ Regulates NO_3_^−^ Uptake via an ABA-Induced Negative Feedback Loop

Soluble receptors of ABA have been thoroughly studied in *Arabidopsis thaliana*. They belong to a family of 14 proteins, named Pyrabactin Resistance/Pyrabactin-Like (PYR/PYL) or Regulatory Components of ABA Receptor (RCAR) and are referred to as PYR/PYL or PYR/PYL/RCAR. Mechanisms of ABA signaling pathway start with the structural changes in the PYR/PYL receptors induced by binding the phytohormone. Changes in the structure of the receptors allow them to sequester members of the clade A negative-regulating protein phosphatase 2Cs (PP2Cs) such as the phosphatase ABA insensitive (ABI2) [37,38]. Consequently, targets of ABI2 such as Ca^2+^ sensor/kinase complexes, CBL/CIPK, are not dephosphorylated by the phosphatase which affect their activity. Precise activation of CBL/CIPK complexes in order to phosphorylate their target proteins often requires CIPK autophosphorylation and CIPK-dependent phosphorylation of the Ca^2+^-sensor moiety in the associated CBL [39]. The interactions between CBL1, CIPK23, AtNPF6.3, and ABI2 were shown unequivocally *in planta* by bimolecular fluorescence complementation (BiFC) [26]. In vitro phosphorylation assays showed that CIPK23 autophosphorylation and CIPK23-dependent phosphorylation of CBL1 was dramatically lowered in the presence of ABI2, thus revealing that ABI2 effectively dephosphorylated CBL1 and CIPK23 (see Figure 4A in [26]).

In the presence of high exogenous NO_3_^−^, a gradual increase in bioactive ABA content in the root is triggered as mentioned above, and the latter is recognized by the receptor PYL/PYR, which after structural change sequesters the phosphatase ABI2. As a result, CBL1, CBL9, and CIPK23 complexes remain active and able to phosphorylate their targets, because these complexes cannot be dephosphorylated by ABI2.

Thus, ABA appears as a determinant in the regulation of NO_3_^−^ uptake by NO_3_^−^ itself through a phosphorylation process by recruiting the CBL1, CBL9 and CIPK23 complexes and actually acting as a signaling module responsible of the phosphorylation of AtNPF6.3, causing a decrease in its NO_3_^−^ absorption capacity. This process has been demonstrated by Harris et al. [40] and has been described as a slow-acting negative feedback loop, activated by NO_3_^−^ itself (Figure 2).

## 5. ABA Functionally Links NO_3_^−^ and K^+^ Uptake via the Action of CBL/CIPK

The signaling module constituted by CBL1, CBL9, and CIPK23 is also involved in the regulation of K^+^ uptake by regulating the cognate transporter, AtKAT1. This finding is supported by data obtained by omics-based techniques and molecular physiology approaches that have improved our understanding of the management of K^+^ acquisition by plants. At least 7 of the 9 members of the Shaker family of Arabidopsis were characterized in heterologous systems (e.g., Xenopus oocytes, insect and mammalian cell lines, and yeast) and found to be highly selective K^+^ channels with various rectification properties, i.e., inward, outward, and weakly inward. The functional data obtained in heterologous systems along with subcellular localization and characterization of a knockout mutant (*akt1*) showed that the inward rectifier AtAKT1 is the major K^+^-uptake system in the root, and is present in the root cortex, epidermis, and root hair [42]. In the same way as AtNPF6.3, AtAKT1 was found to be regulated by a calcium-dependent signaling pathway involving sensors of CBLs family and a target kinase [32,43,44]. A precise combination of a genetic approach (yeast two-hybrid) and electrophysiology (Xenopus oocyte patch-clamping) showed that CBL1 and CBL9 and their target kinase (CIPK23) form alternative complexes located at the plasma membrane where they jointly activate the AtAKT1 channel, thereby increasing K^+^ uptake capacity [43,45,46]. These experiments present evidence that CBL1/9-CIPK23 physically interact with AKT1 channel protein to switch on its activity by phosphorylation [47,48].

Interestingly, one can observe that when the availability of NO_3_^−^ is high, AtNPF6.3 is negatively regulated with a decrease in the influx of NO_3_^−^ via an ABA-mediated phosphorylation process, and concomitantly, AtKAT1 activity is upregulated leading to an increase in the influx of K^+^ (Figure 3). Moreover, it has been shown that the regulation of K^+^ loading in the xylem and its translocation to the shoots was also dependent on NO_3_^−^ availability. More precisely, one could observe that the K^+^-mediated translocation to the shoot through the xylem-loading K^+^ channel SKOR (Stelar K^+^ Outward Rectifier) is stimulated in the presence of high amounts of NO_3_^−^ [47,49]; consistently, expression of *AtSKOR* was upregulated by high NO_3_^−^ (5 times higher under 10 mM compared to 1 mM) in roots of Arabidopsis Col-0 irrespective of the level of K^+^ supply (see Supplemental Figure S9 in [50]).

Furthermore, the link between these two ions is supported by the long-lasting observation that in most plant species K^+^ uptake from the soil is positively correlated with NO_3_^−^ uptake [47,51,52]. This effect was explained at the whole plant physiology level taking into account charge balance in plant tissue, in particular xylem, where K^+^ would serve as counterion to compensate for the negative charge of NO_3_^−^ [53,54,55]. This is very well illustrated by Engels and Marschner [56] who shown in maize that the K^+^ flux rate (µmol/h/g root) was 30% higher under nitrate compared to that under ammonium nutrition. Consequently, K^+^ absorption and accumulation would increase when NO_3_^−^ availability and absorption increase.

**Figure 3 ijms-23-15631-f003:**
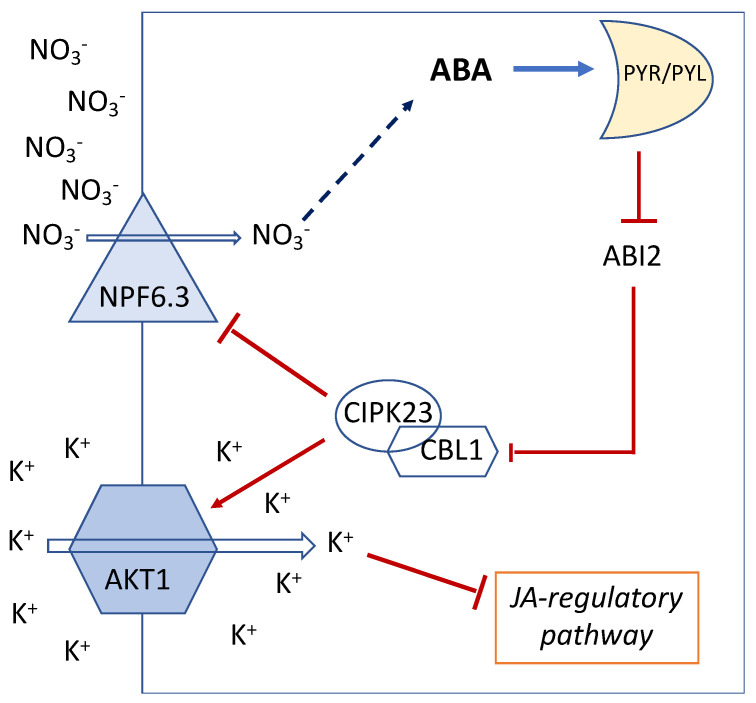
Schematic representation of functional links between NO_3_^−^ and K^+^ uptake via the action of CBL/CIPK complex. Under high levels of exogenous NO_3_^−^, the CIPK23/CBL1 complex is set free to phosphorylate AtNPF6.3 and AtKAT1 thus leading to a decrease in NO_3_^−^ influx and oppositely an increase in K^+^ influx. It is proposed that the increased K^+^ concentration inhibits JA-regulatory pathway of defense against pathogens. The latter is known to be up-regulated during K^+^ deficiency and inhibited upon K^+^ resupply [57,58,59,60,61].

## 6. Jasmonic Acid Links K^+^ Content and Defense against Necrotrophic Fungi

It has been proposed that necrotrophic pathogens induce JA-dependent defense in plants [62,63,64]. Among several K-dependent changes in metabolites of *Arabidopsis thaliana*, accumulation of indole and aliphatic glucosinolates appeared as a characteristic of K-deficient plants [57]. This finding is in agreement with transcriptome analyses in which expression of genes related to JA biosynthesis was enhanced in K-deficient Arabidopsis plants [58,59]. The gene encoding *AtLOX2*, which catalyzes the first committed step in JA biosynthesis [60,61], responded to low K prior to developmental symptoms, e.g., growth retardation and senescence, demonstrating that the induction of the JA pathway was not a secondary effect of stress symptoms [57]. Furthermore, levels of JA, as well as its precursors 12-oxo-phytodienoic acid (OPDA) and hydroxyl-12-oxo-octadecadienoic acids (HODs), were increased in K-deficient plants [57]. Expression of genes involved in JA signaling and response to biotic stress in particular defense genes dependent on the function of the JA receptor coronatine-insensitive 1 (COI1) was also boosted by K^+^ starvation [58,59].

It has been observed that a low NO_3_^−^ supply to Arabidopsis mutant *nrt1.5-5* caused K^+^ deficiency in the shoot as a result of an impairment of K^+^ loading in the xylem at the root level [50]. Interestingly, the survey by q-RT-PCR of the expression of 34 genes related to JA biosynthesis (e.g., AtLOX2), calcium signaling (e.g., ATCML41), defense (e.g., the JA-induced genes AtPDF1.2b, AtNATA1, and AtTPS4), secondary metabolism, and reactive oxygen species production were all more than twofold up-regulated in the mutant compared to the wild type (see Supplemental Figure S4 in [50]); 26 of the 34 tested genes were reported by Armengaud et al. [58] to be up-regulated by K^+^ starvation. Altogether, these results strengthen the idea that K^+^ status is an important player in plant response to necrotrophic fungi-induced disease by modulating JA synthesis and signaling pathways.

## 7. Conclusions

In the pathosystem *Arabidopsis thaliana*/*Botrytis cinerea*, only a small set of 182 genes exhibited an altered level of expression when NO_3_^−^ is supplied to the plant [9]. Among those encoding proteins involved in plant defense, four genes, *PME7*, *PR1*, *PR5,* and *PDF1.2a,* were selected for a more detailed expression analysis. Interestingly, only the level of expression of *PDF1.2a*, a marker gene of the JA signaling pathway was fourfold higher in infected plants grown in the presence of low amounts of NO_3_^−^. These infected plants did not develop disease symptoms compared to the infected plants grown on high NO_3_^−^ which developed visible symptoms of fungal infection (see Figure 4 in [9]). It is thus concluded that in infected plants when the supply of NO_3_^−^ is high (2 or 10 mM), it counteracts the fungal-induced up-regulation of *PDF1.2a*, thus weakening plant defense against the fungi. Moreover, an NO_3_^−^ supply above a certain threshold could contribute to plant tissue K^+^ enrichment which in turn is not favorable to the induction of the JA-dependent defense pathway (Figure 3). In line with this finding, the JA-dependent defense pathway is known to be up-regulated under K^+^ deficient conditions and inhibited when K^+^ is resupplied to the plant [57,65].

## Figures and Tables

**Figure 1 ijms-23-15631-f001:**
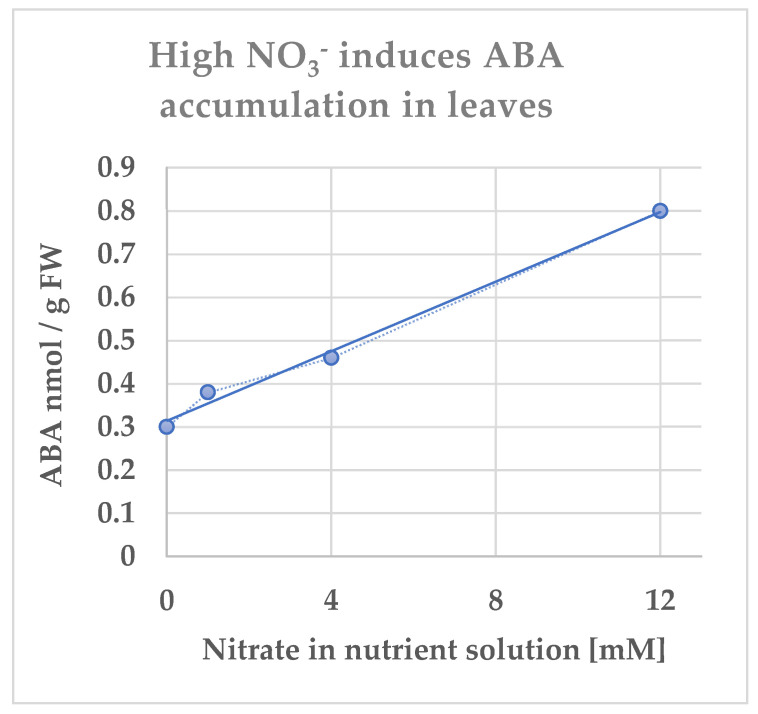
Concentration of ABA per fresh weight of leaves of *Ricinus communis* grown under different nitrate concentrations 41 d after sowing. Data are from A. Peuke (Personal communication), see also Peuke et al. [34].

**Figure 2 ijms-23-15631-f002:**
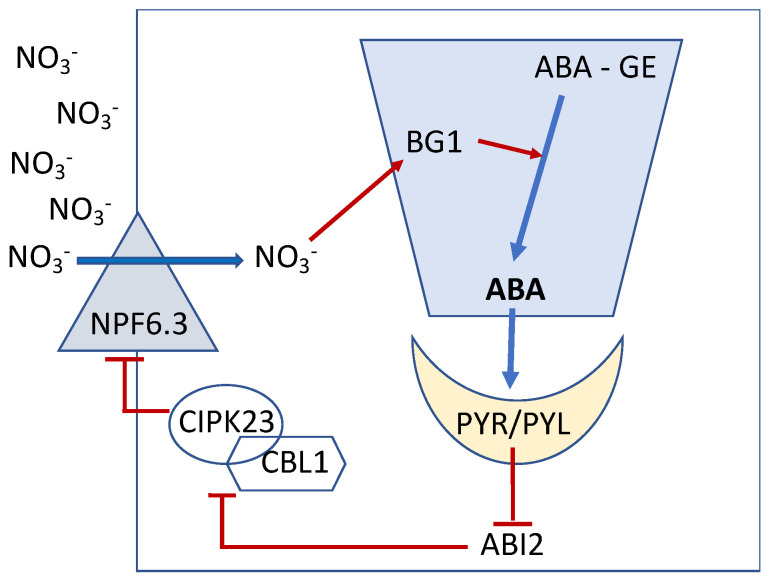
Overview of NO_3_^−^ signaling via the release of bioactive ABA in the Arabidopsis root tip. Nitrate is absorbed through the transporter and sensor AtNPF6.3 (AtNRT1.1). Under high levels of exogenous NO_3_^−^, this ion, once absorbed, rapidly stimulates the expression of the *AtBG1* gene, encoding an ER-localized β-glucosidase, that cleaves the inactive ABA conjugate, ABA-glucose ester (ABA-GE), releasing bioactive ABA. The increase in ABA levels is a slow and gradual process during which ABA binds the intracellular PYR/PYL receptor, causing the inactivation of the ABA co-receptor, ABI2. Once ABI2 is inactivated, the CIPK23/CBL1 complex is free to phosphorylate AtNPF6.3, thus inhibiting its ability to transport NO_3_^−^. Schematic presentation modified from articles of Ondzighi–Assoume et al. and Harris [36,41].

## Data Availability

Not applicable.

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
