# Peer review of "Does Potassium (K+) Contribute to High-Nitrate (NO3) Weakening of a Plant’s Defense System against Necrotrophic Fungi?"

_ijms, 2022, doi:10.3390/ijms232415631_

Round 1

Reviewer 1 Report

the article is interesting, but needs some changes and reinforcement with more recent references.

Author Response

Answers to Reviewer 1

General comment : the article is interesting, but needs some changes and reinforcement with more recent references. 

Answer :

Remark 1 : rephrase the sentence, the beginning of the sentence is a little confusing.

Answer 1 : The sentence was rephrased. In the modified manuscript see from lines 41 to 42.

Remark 2 : change as suggested by the authors .... (p2, line 62).

Answer 2 : The sentence was changed in order to strictly stick to the scope of the two articles to which it refers. In the modified manuscript see from lines 60 to 63.

Remark 3 : little confusing look like ideas separated by short sentences (p2, lines 82 to 87).

Answer 3 : Changes were done according to reviewers remark, in particular, two short sentences were fused into one sentence. In the modified manuscript see from lines 83 to 89.

Remark 4 : it makes no sense to be a review article and have analysis of results, with a graph that is not owned by any of the authors.

When using images from other authors, you must have permission to do so.

However, at this point, they could explain the main aspects without being analysis of results, highlighting the most important.

Answer 4 : These data, although unpublished represent an opportunity to show a clear positive correlation between the leaf ABA content and the amount of nitrate supplied to the plant. The data were communicated by Prof. Andreas Peuke who allowed us to use them for publication in the form of a figure (see the acknowledgments). Moreover, a successful collaboration with Prof. Andreas Peuke has already been concretized though a common publication (Cui et al. Plant Cell Env 2021).

Remark 5 : The same thing for all the figures.

Answer 5 : Figures 2 and 3 are original and were never published earlier by any of the three authors. For these figures we were inspired by those published by Ondzighi-Assounme et al., 2016 and Harris 2017.  Therefore, this was mentioned in the figure legend indicating that our schematic presentation is a modification of the figures published by these authors.

Remark 6 : Reduce the conclusion, highlight the more important aspects.

Answer 6 : Conclusion has been modified in order to implement reviewer’s request and substantially shortened (p8,  lines 254 to 268).  

Remark 7 : Need more recent articles for reforcing the study.

Answer 7 : Recent references were added : references 17, 18 and 21.

Reviewer 2 Report

Dear Authors,

Interesting work systematizing information on the interactions between K, NO3- and the reaction of plants to biotic stresses - necrotrophic fungi. Well-documented information from literature mainly from the last 10-15 years is read with interest.

In conclusion, a good summary of the direction of possible research to expand knowledge on the relationship between NO3 signaling and JA.

The only remark is that the majority of the review is based on studies with Arabidopsis thaliana (except examples for maize and Ricinus communis). The title says "plant" so it would be good to refer to other species as well.

In addition, note: please explain all the abbreviations used in the place where they are used for the first time, e.g. in the ABA abstract.

Author Response

Answers to Reviewer 2 :

The only remark is that the majority of the review is based on studies with Arabidopsis thaliana (except examples for maize and Ricinus communis). The title says "plant" so it would be good to refer to other species as well.

Answer : we agree with the reviewer, we have enriched the article with works on tomato and rice. Still, the majority of the available literature, dealing with nitrate as a nutrient and signal molecule is based only works carried out on Arabidopsis.

In addition, note: please explain all the abbreviations used in the place where they are used for the first time, e.g. in the ABA abstract.

Answer : done.

Reviewer 3 Report

Dear Authors,

i find your ms well written. My concern it is at the conclusion section, i dont understand enough the connection between high-NO3- acts via K+ signaling on JA pathway. I belive you must check this point again. Also in this section of your ms you only propose (lines 255-257, 259-263). You must add something to the reader understand what you propose.

Author Response

General comment : I find your ms well written. My concern it is at the conclusion section, i dont understand enough the connection between high-NO3- acts via K+ signaling on JA pathway. I believe you must check this point again. Also in this section of your ms you only propose (lines 255-257, 259-263). You must add something to the reader understand what you propose. 

Answer to the general comment : The conclusion has been substantially shortened and focused on the main take home message of the article.

Remark 1 : reference is missing (p2, line 59).

Answer 1 : We have added a reference (p2, line 60 in revised ms).

Remark 2 : please add full classification (p3, line 119 seems to pertain to the abbreviation ABA at the beginning of a sentence).

Answer 2 : We have written abscisic acid (ABA) so that we are not stating a sentence with an abbreviation.

Remark 3 : better quality for Figure 1.

Answer 3 : A figure of better quality is uploaded on submission website.

Remark 4 : Check please (p4, line 137).

Answer 4 : We have checked and PYRABACTIN RESISTANCE/PYRABACTIN LIKE (PYR/PYL) or REGULATORY COMPONENTS OF ABA RECEPTOR (RCAR) is correct.